# Reality Capture in Construction Project Management: A Review of Opportunities and Challenges

**Godfred Fobiri \*, Innocent Musonda**  **and Franco Muleya**

Center of Applied Research and Innovation in the Built Environment (CARINBE), Faculty of Engineering and the Built Environment, University of Johannesburg, Johannesburg 2028, South Africa
\* Correspondence: 220158469@student.uj.ac.za

**Abstract:** Reality Capture (RC) is a state-of-the-art technology for digital data gathering and visualization of the actual environment through virtual means. In recent years, RC has contributed significantly to the digitalization of the construction industry globally. However, there are no systematic critical analyses of the benefits and challenges of RC technologies in construction project management (CPM) to drive its adoption. This research provides a detailed overview of the potential benefits and constraints of RC to tackle CPM concerns successfully and efficiently. This study uses the PRISMA procedure to conduct a systematic literature review. Based on the inclusion and exclusion criteria set for the study, 96 articles were considered relevant for review. The articles were analyzed using content analysis techniques to synthesize identify emerging themes. A four-step procedure was used to classify the articles into pre-construction planning, designing and construction. The review show that (a) RC is useful during the planning and designing stage, as the success of a construction project depends on accurate data to reduce the risk of cost and time overruns; (b) the benefits of RC adoption are accurate data, reduced time spent on project monitoring, progress tracking, and quality assessment; (c) RC on a construction site aids in the resolution of the industry's fragmented nature through collaboration; quick and prompt decision making through remote monitoring and control of projects. RC is used as a visualization tool and for digital documentation of as-built models, construction verification, and flaw discovery, thereby improving work processes to achieve project success; (d) the most challenging aspect of incorporating RC on a construction site is the high investment cost. In the digital revolution era, this study could aid in the knowledge and optimal use of Reality Capture in numerous areas of CPM.

**Keywords:** Reality Capture; 3D laser scanning; photogrammetry; construction management; construction project management

## 1. Introduction

The construction industry has seen enormous development and improvement all over the world in terms of digital transformation. The construction industry has altered methods and strategies by building massive and greater infrastructure projects. Similarly, the construction industry has used technology to improve employees and retain new people and procedures in the building occupation to solve labor shortages, long-duration and defective work, inefficiencies, poor quality, and cost overruns [1,2]. Among the technologies currently used in construction project management is Reality Capture (RC). RC is a quick and effective method of producing a 3D dataset (point clouds or meshes) from project site conditions utilizes laser scanners and photogrammetry, either hand-held, with a tripod, or on an unmanned aerial vehicle (UAV) [3]. RC is a system that improves project efficiency, accuracy, value, and safety. Building geometry, construction typology, and material quantities are all included in the 3D models. The 3D models are created from digital data obtained from Reality Capture. RC provides a significant advantage over traditional construction data acquisition methods [4]. The RC technology allows for the replication of

the physical world into a virtual environment, allowing Architectural, Engineering, and Construction (AEC) experts to easily plan, monitor progress, and compare as-built models to as-designed models, ensuring quality control [5]. According to Fobiri et al. [6], due to its practical capacity to collect digital data, RC is among the driving forces behind the development of Construction 4.0. Hence, RC is a technology that could help the building industry embrace the digital revolution through the acquisition of digital data.

Laser scanning is gaining appeal in the sectors of architecture, engineering, and construction as a tool to organize space and transform the built environment. Laser scanning provides technological help for sustainable development goals because of its efficiency, high accuracy and precision, minimal time consumption, safety, and non-invasiveness [7]. There have been a variety of applications of RC in the AEC industry recently; RC has been used in various studies to improve operations and facility management by digitizing heritage structures and monuments [2,8–10]. To produce an accurate and full 3D representation of the monuments, Luhmann et al. [11] recommended combining UAV photogrammetry and laser scanning data.

Researchers claim that RC has proven to be useful in assisting construction managers with tracking and monitoring construction sites [12–15]. Bosché et al. [16] used a 3D laser scanner to trace the construction progress scheduling. Omar and Nehdi [17] proposed using range images from photogrammetry to measure BIM 4D deviation from actual planning. Almukhtar et al. [1] used a proof-of-concept approach to investigate the data acquisition phase of using 3D laser scanners to gather and process data of construction projects. For monitoring multi-building development sites, Nath et al. [18] proposed a fully automated Reality Capture method. The research looked into some of the unique challenges that arise while aligning multi-building point clouds. To monitor multi-building construction sites, Masood [2] presented a completely automated three-step alignment approach. Assessing quality can be monitored with RC and point clouds to create as-built 3D models [19–21]. Hamledari et al. [22] proposed a reliable method for updating as-designed BIM-based on-site observations to reviewed building components using the industry foundation classes (IFC) schema. Reality Capture, according to Murphy et al. [23], helped the predictability of higher output and quality in prefabrication by incorporating existing building components into 3D models built from 3D scans. Hamledari and Fischer [24] described a simple contract-based system combining blockchain and Reality Capture for autonomous building progress payment administration. Cribbs [25] highlighted the relevance of Reality Capture and embraced it as part of the ideal state workflow model for increasing labor time utilization for BIM-based prefabrication. In a pandemic condition, Mchugh et al. [26] evaluated the effectiveness of the RC digital approach. They described how the RC strategy helps to counteract the divided teams produced by the COVID-19 epidemic by increasing stakeholder participation and collaboration on building projects.

RC has been successfully used in a number of sectors in the building industry with a wide range of applications. It can help a project succeed by accelerating accurate digital data acquisition for design development and communication with all parties involved, from the owner to the laborer, exceeding owner expectations, and lowering project costs through efficient project delivery. In the field of construction management, RC technologies are being used to put a 3D model visualization in front of clients, consultants, and contractors, stimulating the experience and knowledge of design concepts like never before [27]. Professionals can interact with the model in this way and deal with faults, structural assessment, constructability difficulties, risks, and cost before they exist. RC is a unique learning opportunity for clients by allowing them to identify and visualize a project remotely in a safe, hazard-free environment in real time. RC technologies are utilized for construction progress monitoring evaluation, defects management, quality management, project scheduling, accurate digital information collecting, safety management, logistics management, facility management, and so on.

There is no summarized information regarding the benefits and constraints of employing RC in construction management available in a single study in the literature. The

motivation is to review and organize the opportunities and challenges to drive its continuous adoption by providing a one-stop solution for prospective researchers and practitioners.

This study is aimed at providing potential readers with practical understanding of RC usage. As a result, the goal of this research is to determine the efficacy, application potential, and difficulties of RC technologies in construction project management. This research could aid authorities in better understanding the usability and efficacy of various RC technology and applications in solving a variety of construction management difficulties.

## 2. Research Methods

In recent years, RC technologies have been widely used in construction project management, either for research or as an experiment. Quality management, planning and scheduling, defect discovery, and progress management are all areas where different researchers have had success. As a result, the use of RC and its opportunities and benefits in construction management are widely discussed in the literature. Without a summary organized with information, knowing and learning all of the optimal use of opportunities in numerous areas of construction management is quite problematic. This research purposed to offer a comprehensive overview of RC technologies, as well as identify the significant challenges and opportunities these technologies bring to the built environment sector. A systematic literature review was conducted to accomplish the aim using the PRISMA (Preferred Reporting Items for Systematic Reviews and Meta-Analyses) procedure to ensure replicability [28]. Academic peer-reviewed articles were used to compile the RC literature. PRISMA is a protocol established for conducting a comprehensive literature review of a four-phase flow diagram [29]. PRISMA was chosen over alternative protocols because of its widespread acceptance, comprehensiveness, and wide variety of applications in numerous academic disciplines, as well as its ability to improve the accuracy and transparency of academic literature reviews [29,30]. The PRISMA approach has been adopted in a number of investigations in the AEC industry [31–33].

Following the PRISMA protocol, the study adopted a similar three-stage methodological approach [34]: The research objectives that address the study question, keywords, and a set of exclusion and inclusion criteria are included in stage 1 (planning). The goal is to identify the most common RC technologies application areas in the construction industry, as well as the opportunities and challenges RC presents throughout the project, based on RC in the pre-construction, planning/designing, and construction stages. To fulfill the study's goal, a thorough literature evaluation was undertaken using a diversity of sources, together with journal papers, conference papers, books, and book chapters; all were linked to RC in the construction industry. Relevant publications were found by searching the Scopus databases with keywords, as this is the main source of data with a wide range of science, technology, social science, and other subjects offered, and is the most credible science database for effective systematic literature reviews [35]. According to Zhang et al. [35] the Scopus database is the most credible and effective systematic literature review. It offers a high level of accuracy compared to other search engines [36]. There is no double citation in Scopus-indexed journal articles and its document citation is highly reliable. Additionally, it is one of the most significant peer-reviewed sources of journal articles, conference proceedings, peer-reviewed papers, book chapters, etc., covering a considerable quantity of abstracts and citations [37,38]. These keywords specified the study fields and provided an overview of the existing RC technologies in use in the industry. The following keywords were used: "construction" or Construction project" or "Construction project management" AND ("reality capture" OR "photogrammetry" OR "3D laser scan" OR "laser scanning").

In stage 2 (conducting the review stage), relevant articles were searched in May 2022. The use of RC in building has largely gained traction in the last two decades, with adaptation increasing in the preceding decade. The initial search resulted in 3500 articles. To cover a broader scope of the extant literature, initial search criteria were established: all English-only journal articles, conference papers, books, book chapters, and trade journals

related to engineering from 2010–2022. The selected year of publication was greatly influenced by a combination of practical consideration of the development of Reality Capture in construction-related research and the 4IR in the digital transformation era.

The number of relevant articles was reduced to 289 after the initial search criteria. The title, abstract, and keywords of the remaining 289 articles were extensively screened and examined by removing duplicates and unrelated articles to attain a high degree of quality. The importance and significance of the academic literature in the evaluation process were confirmed by carefully checking the abstracts of each article chosen. Eligibility criteria developed to guide the selection of the articles during screening were that the article's title, abstract, keywords, and full text must describe and relate to the review objective. As a result, articles with keywords in the abstracts or titles that failed to describe the research topic in the body of the text were removed. After screening, 96 papers were allowed for further review, while 104 relating to facility management and 89 unrelated articles were eliminated. The 96 articles were considered adequate for the systematic review as indicated in Figure 1.

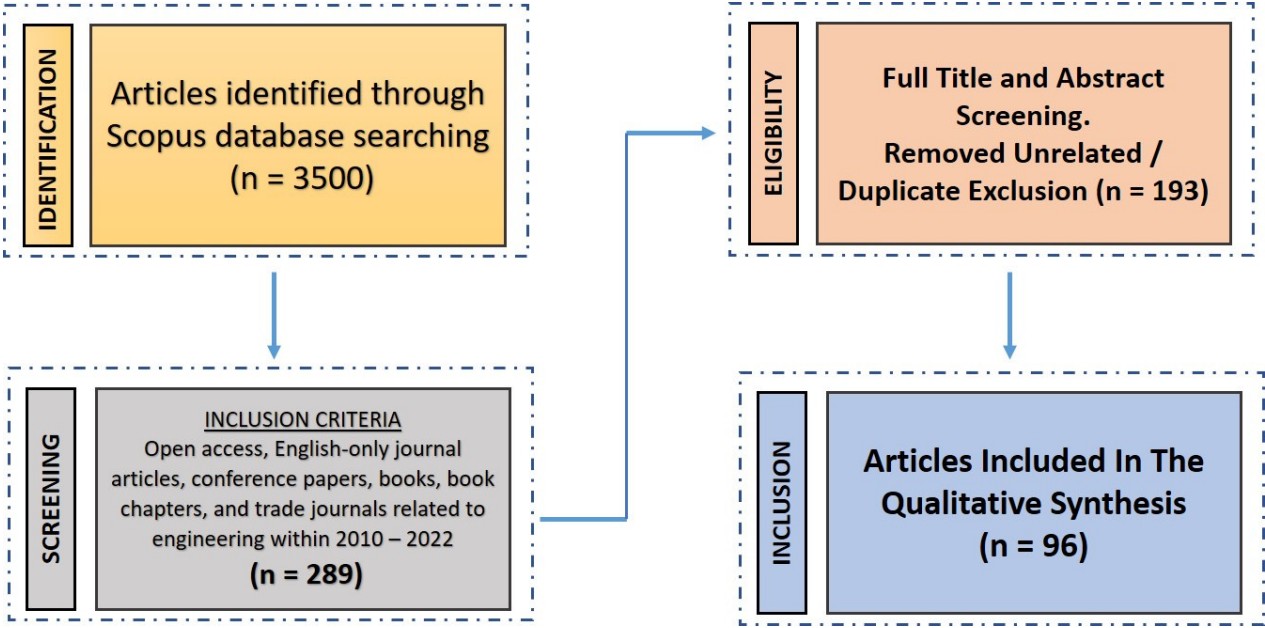

**Figure 1.** PRISMA framework article collection process (n = number of articles).

In stage 3 (reporting), 96 selected articles were analyzed using content analysis methods to synthesize the themes. Each of the articles was carefully read and all relevant information needed to address the main objectives of this paper was documented. To ensure consistency of the extracted criteria, a careful examination of extracted information was conducted. The reviewed literature was then categorized into distinct topics using the four-step procedure. The first phase drew attention to key opportunities and problems in the studied literature of RC in construction management. Secondly, the most essential themes were then grouped and thirdly, reviewed in relation to the study's objectives. The final stage was to compare the categories to those found in earlier reviews to see if there were any new issues. Finally, the themes were grouped and refined under common themes during the pre-construction, planning and design stages, and during construction.

## 3. Results and Discussion

### 3.1. Annual Trend of Publications

Figure 2 reveals the trend of the annual relevant publications. Between 2010 and 2012, only one publication for each year was recorded. In 2013, the number of publications showed an upward trend of nine which indicates that the application of RC is attract-

ing increasing attention from researchers in the AEC industry. This is as a result of the fourth industrial revolution agenda and rapid implementation of digital technologies in construction [39].

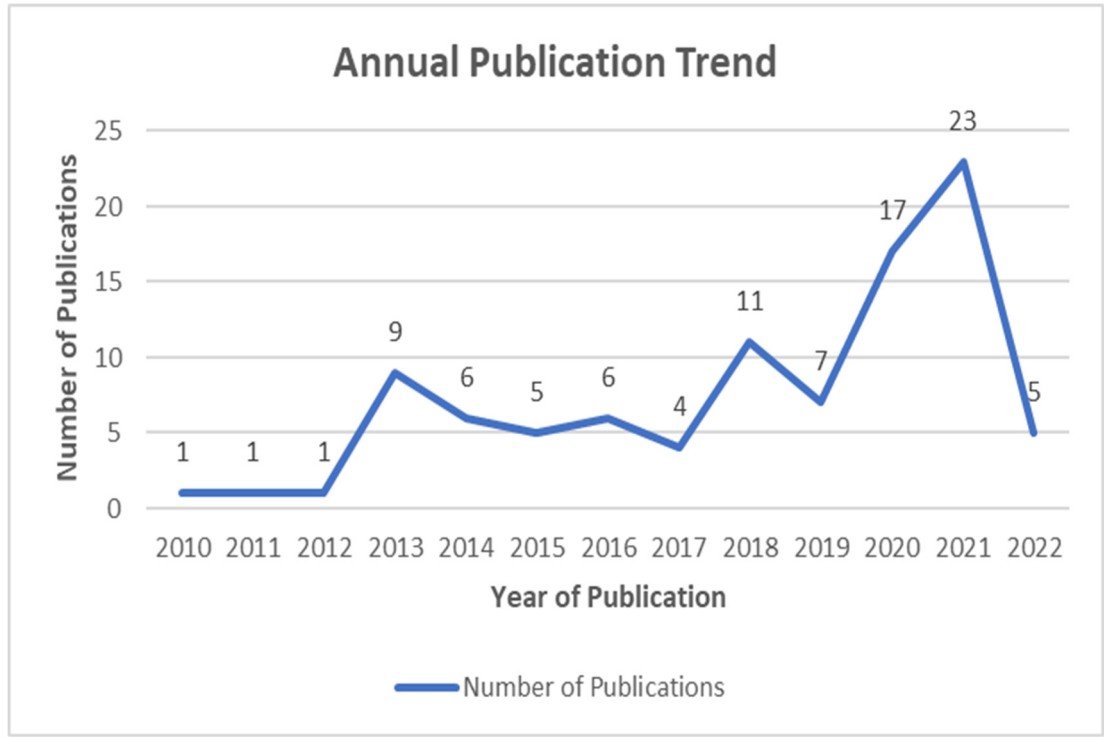

**Figure 2.** Annual trend of relevant publications.

A downward trend was realized from 2014 to 2017 before recording a significant raise in 2020 and 2021 with 17 and 23 publications, respectively. The data point out that the subject is interestingly growing attention with the anticipation that the RC technology adoption in the AEC industry is increasing. However, the journals in the first five months of 2022 exceeded the annual articles published before 2012. There is an indication of a great contribution to the growth of RC research driving the RC adoption in construction management in the digital revolution era.

### 3.2. Opportunities of RC in Construction Project Management

RC has been extensively used for different applications within the built environment. According to Fobiri et al. [6], RC has been extensively applied to digital cultural heritage documentation, accuracy assessment, project monitoring, building surveys, construction site monitoring, and quality control assessment. To effectively implement novel technologies into projects, professionals need to recognize the benefits and performance enhancements RC offers to the built environment. This review offers project professionals a one-stop shop of summarized information on the practical advantages and constraints Reality Capture offers to construction project management delivery. The following review classified the opportunities into pre-construction and construction phases.

### 3.2.1. Pre-Construction Phase—Planning and Designing

Planning is a vital part of the construction process because it determines a project's success in terms of time, cost, quality, and safety [40,41]. Inaccurate planning will lead to project failure [41,42]. The planning stage involves many stakeholders and includes forecasting, scheduling, cost analysis, and risk assessment [43]. These are carried out based on the type and kind of data gathered and available. In the early stages of a project, RC is crucial since it has the ability to acquire quick and accurate digital data.

To increase efficiency and productivity at this level, the commercial sector has made large investments in research and development of new RC-related software [43]. The project planning team merged Reality Capture and real-world data to build context models that can completely resolve the existing built and natural surroundings during the planning and designing stages. Throughout the design stage, conceptual design, analysis, detailing, and documentation are all carried out. In the pre-construction phase, BIM data are used to inform scheduling and logistics. Traditional techniques are streamlined in the design phase by using various automated tools such as AutoCAD Autodesk, Navisworks, Revit, SketchUp, etc., software. In the design phase, data provided by RC photogrammetry drone footage help to create three-dimensional building information modeling (3D BIM) which becomes a reservoir of information. During the phase, RC can assist the project management team with a high level of detailed information on the site condition to aid in planning and designing to reduce the level of project uncertainty. By boosting communication and collaboration among the design team and functions, RC to BIM (scan-to-BIM) will improve the current administration and control in all parts of architectural practices. Incorporating information and knowledge repositories into various initiatives could also make them function more smoothly. This will ensure that all stakeholders have access to updated data and that information can be quickly transferred among starkholders [24].

Data Accuracy and Reliability

Most critical decisions that influence project success are taken at the planning stage. Project stakeholders usually take decisions based on the data gathered from the project site condition, clients brief, and soil condition. Therefore, the level of accuracy and reliability of project information is crucial for the project success, e.g., data inaccuracy, poor design, poor budgeting, and forecasting resulting in time and cost overruns. The project information aids architects in design preparation; structural engineers also rely on the same information for structural analysis and detailing. Quantity surveyors determine the project cost estimation and budget on the same data. Data acquisition carried out manually are saddled with challenges, such as time-consuming, error-prone, and infrequent [13,44], which have the tendency to affect the project outcome. Therefore, data gathered at the planning stage are crucial for entire project management, in quality, time, cost, and resources [45,46]. RC is a quick and effective method of producing a 3D dataset from project site circumstances [3]. Digital data from RC can greatly increase project effectiveness, precision, value, and safety since it consists of vital data such as geospatial data, nature/topography of land, surrounding environment, etc., offering a better advantage than the traditional methods [4]. The 3D model obtained through Reality Capture permits the replication of the physical world into a virtual environment to obtain valuable information for AEC professionals to easily produce project documentation accurately and efficiently [5].

Effectiveness and Efficiency of AEC Professionals

There are inefficiencies in the traditional method of delivery of AEC professionals—in terms of time and output. Design error and time wasted exist when putting together different parts of the blueprints. Several site visits during the designing phase use assumptions on hard-to-reach and difficult-to-measure areas and objects [3] thereby, increasing the level of risk of the project. Quantity surveyors spend lots of time creating quantity take-offs. Traditional methods of building project delivery, which are deemed inefficient and lead to time and expense overruns, are being phased out [47–49]. The situation becomes worse on a large and complex construction and infrastructural project. RC technologies help AEC professionals be more efficient and effective in infrastructural delivery because of the accurate and detailed data used for planning, designing, and decision making [50,51]. RC is employed in the planning phase to make effective and timely decisions. It also aids the consultants and contractors in designing and constructing a project that is feasible to build.

### Site Condition and Design Visualization

The visual mental representations used by the designer during the design process are referred to as visualization during design. Through spatial representation, visualization allows for the generation, interpretation, and manipulation of data [52]. This is greatly enhanced when digital data of the site condition are presented for design. In other words, it is the mental pictures used by a designer when completing a design task that is improved, reducing the design errors and buildability issues. Predictability is improved because of the usage of the 3D models, enabling the assessment of the building and structural components of the design [23,53,54]. As a result, collaborating improves data distribution and reduces uncertainty. This will give parties a clear picture and understanding of the site condition for their input at the design phase of the project. Three-dimensional models provide vital elements to form a basis for other visualization technologies such as virtual, mixed, and augment reality (VR/MR/AR) [55].

### Reduction in Project Risk

Many uncertainties increase the project risk in the implementation phase of construction projects. Unforeseen circumstances can lead to a variety of uncertainties and have a significant impact on the project's performance [56–58]. RC offers an advantage to provide point clouds, a virtual representation of the site condition. Point clouds data give precise, valuable, and detailed information that is difficult to achieve by manual means. Therefore, it is imperative to gather initial data of the site condition using RC, proving precise and more detailed information for the planning and design phase.

### Prevent Cost Overruns

Project cost overrun is a global concern with researchers and practitioners seeking a lasting solution through modern technology [58–61]. Among the identified critical causes of cost overruns is inaccurate design and budget preparation [27,41,62,63]. The design and cost budget preparation largely depend on the level of accuracy of data available for use. Poor and inaccurate project data affect the design and budget cost leading to a lot of project variations in design and cost. Using RC for initial data acquisition goes a long way to prevent cost overruns since the project cloud data give the project team an accurate and high level of details for designing and planning which reduces cost implications. The RC can help the project team predict any unforeseen variation in design and cost based on the project size, contract type, and initial site condition information. This allows the project team to address minor difficulties before they become major problems that might lead to project failure.

### 3.2.2. Construction Phase

During the execution phase, the actual construction is carried out in a practical manner based on the design, requirements, and budget as projected. Project construction logistics are coordinated with trades and contractors to ensure maximum timing and efficiency. The most significant deviations from typical methods are presently occurring during the project's building phase. This portion of the literature examines the benefits of RC to the built environment during the construction phase.

### Data Accessibility

A vast capacity of data is produced during the execution phase of every project; effective project management requires the use of this project data for reporting and decision making. Collecting project data manually for reporting and decision making is subjective to human error and is time-consuming [49,64]. Therefore, the provision of real-time digital data being made available remotely for decision making is imperative. RC has the capability to provide ongoing project data [3,65,66]. However, the request for extremely tailored projects keeps growing, and real-time data access has become critical. Project parties can address difficulties quickly by connecting with numerous project teams and using real-

time data to satisfy changing client expectations [67]. Additionally, RC's availability of real-time site data allows the project team to prioritize concerns and assign a risk score to subcontractors, allowing construction managers to work closely with high-risk teams to minimize dangers [68].

Site Layout Planning in Large-Scale Construction Projects

The arrangement of construction sites is having an impact on construction productivity, safety, and efficiency. Dynamic site layout planning (DSLP) explores the ongoing modification of construction facilities on-site, allowing temporary facilities to be relocated as the project moves forward [69]. In addition, construction sites are reported as an accident-prone industry and are said to have poor productivity [15,70]. The construction industry has the highest fatalities, with over 18% of deadly work-related injuries [71,72]. The site layout planning becomes complex, time-consuming, and confusing when performed manually. To address the issues of productivity and health and safety concerns, there must be an effective and well-planned construction site using the precise and detailed data on an existing site condition. This can be achieved when detailed site information is available; RC can immensely provide such data to plan the site space and layout effectively to avert chaotic site conditions, making the site free from accidents and improving productivity. Hammad et al. [69] undertook research to create a workflow for combining unmanned aerial vehicles (UAVs) and photogrammetry capabilities with the planning of the site pay-out, optimization, and BIM for site layout design automation in major construction projects. The study concluded that numerous characteristics might be examined using data from UAVs, including the closeness to access ways, distance between facilities, and location appropriateness; space optimized and internal transportation cost were gradually lowered.

Monitoring of Site Progress

It is critical to keep track of the advancement on the construction site during the construction period since the projects can benefit by cutting costs, shortening timeframes, and increasing quality. Effective progress control is critical for ensuring that infrastructure building is completed as quickly as possible. Walking across the infrastructure project site to track the progress of various activities takes time and it necessitates the extraction of information from construction designs, timelines, and cost, as well as information gathered on the job site [73]. Insufficient information on the project status results in errors and ineffective measures, causing setbacks and increased expenditures [74]. The use of digital tools to monitor and control projects would lower the chance of error and allow for timely corrective actions. To ensure project execution follows through as per the as-planned, progress monitoring is a key function for project management. This is carried out periodically to prepare a report for stakeholder decision making. This is to ensure that the project is performing according to the project objectives. Project stakeholders' decisions are largely dependent on the project data collected for the report. Therefore, the collection of data that truly reflect the ongoing project in terms of cost, time, quality, and specification, is imperative for stakeholders' decisions. Relatedly, timely reporting is crucial for quick decisions to put the project on track. One of the fields of RC application is construction site monitoring. It provides visual records of accurate data for construction progress monitoring and assessing the actual progress [74–78]. The application of RC for automated project progress monitoring offers incredible support to the project management and the successful completion of the project [3,11,79]. Duarte-Vidal [74] conducted a study on technology integration and interoperability in order to show how they can be used to monitor and control construction projects during the execution phase. The construction progress tracking can be achieved by using UAVs to digitally record the physical progress of a building site and by using point clouds acquired by photogrammetric techniques to record the physical progress of a construction site on a regular basis [54]. The construction sector has a poor track record in terms of productivity, which has been attributed to a lack of effective progress tracking periodically. The majority of existing manual systems do not provide the key

project team with a common comprehension of project performance in real-time, making it difficult to detect any deviation from the original timeline. Using RC technologies, a novel automatic system for monitoring, updating, and controlling construction site activities in real time can be developed. This is conducted by leveraging advances in close-range photogrammetry to deliver an original approach capable of continuous monitoring of construction activities, with progress status determined at any time, all through the construction project life cycle [15].

Enhanced Stakeholders Collaboration

Due to the fragmented nature of the project team, poor coordination and communication between the teams usually exist [26]. Teams often work in isolation with outdated data and plans that are riddled with errors and omissions, having a large effect on project delivery, such as delayed decision making, poor/isolated site instruction, and lack of agreement and clear project direction. Most projects suffer as a result of poor stakeholders' collaboration and communication. The lack of collaboration among AEC teams and the client affects quick decisions in real time. Changes not shared simultaneously and instantaneously brought across the stakeholders delayed decision making [55,80]. The digital data acquisition of ongoing construction projects produced by RC technologies enables and promotes new forms of information sharing, collaboration, communication, and accessibility [26,81–83].

Remote Construction Supervision and Control

Construction sites can be monitored and controlled remotely by gathering data from RC strategically positioned across the site. Data collected can include the stage of completion, location of raw materials, and quality of work, i.e., structural integrity of buildings. This is advantageous because it improves efficiency since the project team can monitor, give instruction, and control the direction of the execution without necessarily being at the construction site all the time [54].

Health and Safety Assessment

The construction industry is regarded as a high-risk occupation with an unwillingness to adapt. Human mistakes, dangerous job activities, equipment malfunctioning, and unsafe work environments result in construction site accidents. Moreover, accidents results and contribute to project delays and cost overruns. Using visual site condition analytics can lower the likelihood of workplace accidents. In comparison to other industries, the construction business has a high rate of injuries. This is because construction workers are frequently exposed to hazards on the job site, such as heights, falling objects, equipment, tools, and poisonous compounds [72]. As a result, construction businesses must adopt a practical method by utilizing RC to lower the likelihood of accidents occurring and teach workers how to avoid them. Sensor-based and wearable technologies for safety monitoring can also support RC to improve safety. Assessment of the digital data can help detect dangers on the job site and alert supervisors for prompt measures. The site data can be analyzed to assess project progress, safety, quality of work; and can help improve their health and safety. The acquired site data, can also be used to identify safety issues and images can be compared to accident data. When there is an impending threat on-site, project managers will be notified to hold a safety briefing. On-site accidents have a negative influence on personnel's well-being, as well as project cost and timeliness [84,85]. Because of the distinctiveness and nature of operations, with the complexity of the work environment especially, high-rise construction projects are an unsafe profession. BIM and other digital technologies such as RC have been highlighted as useful tools for improving productivity, efficiency, and safety of the construction projects [86,87]. To improve construction health and safety management, several developing RC, such as laser scanning and photogrammetry, can be applied. Other emerging technologies and interventions for construction site safety are evolving [88,89].

Project Team Communication and Data Acquisition

Effective communication and information retrieval from the construction site are critical components of a successful construction project [90,91]. In the study, Pejoska et al. [90] initiates that the inclusion of RC technologies considerably enhances access to project data on the job site and effective communication when equated to more outdated data sources. RC technologies are used in the construction sector to collect field data and communicate it to stakeholders. RC enables rapid and easy access to data, allowing project stakeholders to determine remedial actions to save cost and time overruns instigated by performance mismatches. Several businesses are attempting to come out with a light weight tool, according to McHugh [26], to reduce the issues and difficulty of on-site data recovery. Due to increased visual benefits of RC technologies, parties can communicate more effectively while making comments and proposals for a given project. The presented visual capabilities and opportunities of RC technologies allow for increased communication among different parties participating in the building process when observing and giving comments and judgements for a given project phase. RC is among the successful means of gathering site data [2,17,65,91].

Enhance Assessment of Work Done and Approval for Payment

The current approaches to valuing work done and decision making, which is based on manual and paper reports, are often time consuming and challenging. Work done during the execution phase of the project requires a periodic assessment to determine the amount of financial resources invested into the project to date for certification and the reimbursement of the contractor. Usually, the joint measurement and assessment of work done are carried out by the client's quantity surveyor and contractors' quantity surveyor. The traditional paper based approaches are cumbersome and time-consuming, subjective and often results in claims and disputes [41]. By employing RC technologies and retrieving data through dependable automated methods, the analysis improves efficiency, accuracy, and quality [92]. Construction processes will be more dependable, transparent, productive, and efficient thanks to RC which has the capability of offering detailed digital data for visual inspection and assessment by all stakeholders. Calculating the exact volume of the earth is critical. The standard method of calculating the earth's volume necessitates the use of measurements to gather a huge amount of information, which RC can be adopted for [93]. The study by Nguyen [50] presented practical solutions to increase the correctness and effectiveness of the process of quantity management by exploiting recent technical breakthroughs of the 4.0 industrial revolution and employing the opportunities and characteristics of BIM and RC technology. This lessens the time for contractors' claim approvals since the evidence of work done can be readily accessible remotely. It is an intelligent method that automates payment procedures and lowers the potential delay from on-site construction processes [94].

As-Built Model Visualization

A few decades ago, developing a 3D digital model for a construction project before the commencement of a real project was a challenge, let alone for as-built models. However, a 3D project model can now be created employing RC technology that gives the impression of being in the actual world, before the project is visited physically. The visualization of a project as-built model incorporates numerous parametric data previously unavailable in a two-dimensional model. The model serves as a single source of data for all construction sections [52]. Remote control technologies enable a person to remotely access the complete project, including its interior and exterior areas [95]. RC is changing how data is acquired and enable safety improvement, better communication with all project participants, design verification, and can be used to choose appropriate methodologies to address accessibility and space challenges on construction sites [54]. Following the building phase, RC technologies reduce facility administration and maintenance time and cost since better understanding and efficiency, increased productivity, and effective communication is achieved through the ability to visualize the construction activities. RC applications offer a

collaborating, three-dimensional, instantaneous platform for visualizing the project model to promote high-quality, high-safety, and defect-free construction projects [47,54,96]. TLS measurements in the form of 3D point cloud data can be used to virtualize the structure's "as-built" design in steel-framed buildings [97].

RC Gives Life to Building Information Modeling (BIM)

In the design, engineering, and construction industries, BIM is rapidly growing. This is because of RC technology, which has evolved into one of the most important components of BIM, allowing for the capture of semantically rich geometric representations of 3D models in the form of point clouds [98]. BIM is an intelligent 3D model-based method that gives AEC professionals the needed insight and data to effectively design, build, and operate buildings. Architects make use of BIM to produce 3D models including information on physical and functional properties [59]. AEC practitioners may collaborate on synchronized models using BIM, giving everybody a maximum comprehension of their work aligned with the overall project allowing for efficient professional delivery. The data contained in the model establish behavior and connection between the components and the design element [13,99]. When model elements are altered, all the views are updated automatically in the floor plans, elevations, sections, and schedules. The data in the model help gain faster approval for payment since there is a realistic visualization of the intent design on the construction field. BIM increases the effectiveness and productivity of the construction phase by offering insight into a design's constructability, as well as a better understanding of the building's future operations and maintenance [100,101].

However, during the construction phase, there are possibilities of variations in design. The BIM model, therefore, requires an update from the actual construction. Without an update, it is considered a dead BIM because it does not reflect the true information of the ongoing project. The provision of accurate digital data for the BIM update is critical for successful project implementation, operation, and maintenance. The demand for accurate up-to-date digital data in the BIM update to improve the workflow and collaboration among AEC professionals [3] requires the use of RC technology. RC provides quick and efficient digital data of ongoing construction site conditions [3]. The 3D point clouds generated consist of vital data such as building geometry, stage of completion, construction typology, and material properties for BIM processes, offering a better advantage than the traditional construction methods [4]. This enables a scan-to-BIM workflow to be achieved for BIM analytics by evaluating the project performance in terms of cost, time, quality, and safety through the use of various BIM analytics software. BIM is a platform for creating and managing building data about a construction project throughout its lifecycle. The project lifecycle of a building also contains a stage of renovation, and terrestrial laser scanning is an efficient way to develop 3D BIM in their current state. Laser scanning is a non-contact method of capturing the shape of physical things and providing a precise depiction of the building geometry [102].

Scheduling and Cost Monitoring

The usage of RC application models is essential for keeping track of construction costs and schedules. Cost and time forecasts that are wrong have major financial ramifications on the project. The use of RC enables the updating of the combination of 4D and 5D BIM visualization, reducing the risk of unanticipated project cost and milestones [56]. This aids prompt stakeholder decision making on the remedial actions taken to keep the project cost and schedules on track for the project's successful completion.

Productivity Measurement

Because of the increasing complexity of today's large-scale building projects, schedules are very susceptible to delays. The earthwork processes are critical in underground construction sites. Combining two computer vision-based technologies, photogrammetry analysis can be used to evaluate the productivity of soil removal. Throughout the project

execution, RC device photogrammetry is utilized to build periodic point clouds for the assessment of the volume of work done to determinate productivity [103].

Defect and Quality Assessment

Quality assessment was an expensive and laborious issue before the use of RC technology in construction. The flaw is sometimes overlooked, and the report is lost or ruined [104]. However, defect assessment becomes much easier and more successful with the help of RC. There are no chances of missing or failing to disclose damages when using the visualization capabilities of RC [92]. This operation does not require any physical labor. This defect and quality control system saves labor, money, and time [64]. RC is more effectively applied in construction quality assessment and building defect management than before. The as-built models produced by RC can be compared with the as-planned model for quality assessment and construction verification using a BIM analytics software. BIM is a big step forward in the digitalization of the construction process. The virtual BIM model is a repository of graphic data and other information that may be used to verify the geometry of a building's structures. TLS close-range photogrammetry is the most efficient approach for collecting spatial data [105]. A study describes a method for examining the prefabricated wall panels using 3D laser scanning. After panel installation, a standard 3D laser scanner BLK 360 was used to acquire as-built data. Geometric metrics such as angles and lengths can be used to determine whether or not an installation satisfies the quality standard [106]. Quality assessment of complicated geometric shapes of the façade can be achieved by conducting an evaluation of as-built and as-designed panels using RC [107].

Offsite manufacturing (OSM) geometric compliance is critical for assuring proper fit-up, structural integrity, building system, performance, and assembly alignment on-site. A geometric digital twin (gDT) created through 3D scanning can be used to digitize an assembly to discover and fix any difficulties in advance [108]. Quality control is essential to a modular building project's success, and it should be enhanced at every level of the process, from design to construction and installation. Existing methods for determining the assembly quality of a detachable floodwall, which are time-consuming and costly, rely mainly on traditional inspection and contact-type metrics [55]. The construction quality monitoring of blocks is critical in the construction of buildings. The total station device is widely used in traditional applications, but it has various disadvantages, such as a delayed measurement time, a high labor intensity, and a restricted amount of data points. The RC device terrestrial Laser Scanning (TLS) can be used to obtain efficient and precise hull block construction information [109]. In the AEC business, evaluating the quality of construction for conformance with the design purpose has proven to be a difficult process. Data capture techniques, such as 3D laser scanning and photogrammetry, have been used to create 3D as-built models for construction quality assessment. To execute the quality control assessment, the prepared as-built models and as-design BIM were then combined and changed in the Autodesk Navisworks environment. RC technologies have a lot of potential for assessing construction quality [110].

As-Built Digital Documentation

RC ensures up to date as-built documentation. The as-built documentation is critical to generating an as-built BIM model which is carried over to the operations and maintenance of completed assets. This information is also useful when a structure will need to be decommissioned or demolished [23,107,111,112]. The as-built 3D digital model contains objects, structures, and asset information for the operation, repairs, refurbishment, and maintenance of facilities by assessing the status of buildings. RC technologies are becoming increasingly prominent in the maintenance and measurement of infrastructure projects [113–115].

Reduction in Variations

Many difficulties confront the worldwide construction industry, including repeated cost and time overruns, which are exacerbated by deviations that occur during the con-

struction process. Variation is one of the most controversial issues in construction contracts. Modernizing the sector using developing technology has been shown to reduce variances and provide other advantages. The effects of developing technology in reducing the incidence of deviations in building projects are investigated in a study by [116]. When properly implemented, emerging RC technology, such as drones, 3D Laser Scanning, and photogrammetry, as well as BIM, Virtual Reality (VR), and others, are proven to be valuable aids in preventing the development of difference.

### 3.3. Challenges of Reality Capture in Construction Management

Like many beneficial technologies, there are always inherent difficulties in addition to its potential and uses. The usage of RC presents a number of financial, technical, and cultural issues. The scale, people, and capital intensity of the project, the industrial sector, project technologies, and the types of businesses that use the technology all affect technology usage [117]. The primary difficulties that need to be addressed in order for RC to obtain widespread adoption and effective use are workflow, data quality, scan planning, and data processing. Finally, future research options are indicated, including (1) hardware and software cost control, (2) data processing capability development, (3) automatic scan planning, and (4) digital technology integration [118].

#### 3.3.1. Higher Initial Investment Costs

The advantages of RC on a construction site are apparent. However, RC methods have a significant initial cost of investment for procuring the RC equipment, software, and training. The bulk of subcontractors and small enterprises, which make up the construction industry, may not be able to pay. The huge initial costs need a major financial commitment as well as application goal, and these investments will be more exposed to risk, particularly in a highly competitive market. This is a critical challenge for emerging economics to adoption and implementation of RC. Investing in system-level technology is out of reach for small- and medium-sized organizations [77]. Furthermore, because many RC systems users are not fully matured and require continual investment to stay up with technological advancements, a high cost of ownership and use are required. As a result, construction companies must assess the cost savings that RC could bring to a project and determine whether it is realistic. As RC in construction expands and turns out to be more common, the cost is likely to decrease, making it more accessible to small enterprises [48,54,77,110,112].

#### 3.3.2. Technical Requirement

Close-range of RC is delicate and high-tech, and it necessitates a lot of technology requirements for its operation and handling. If the device fails to fulfill the standards, the measurement findings will be distorted, affecting the final results [119]. Extracting meaningful data necessitates the use of contemporary technical instruments [120]. Only by streaming all data to the user live via the Internet with minimum local caching of commonly used data, can all features or functions of digital data be streamed. This means that for basic functioning and improved performance, the user must have a minimum Internet bandwidth. When a large number of individuals is using the same place, it is impractical to employ a network proxy or caching service to reduce network traffic due to proprietary communications protocols [101,121]. A computer system with a powerful processor is required for data processing. Processing time depends on the computation requirement. Large data sets require the use of a specialist computer for data processing [23,92].

#### 3.3.3. Field Operational Difficulties

RC devices are a fast and accurate way to capture point clouds for a variety of applications beneficial to the AEC industry. Construction sites have a plethora of operations involving the mobility of operatives and workers, many of which pose implementation challenges. It is possible that the level of detail in the capture components will be lowered. Before starting the capture date, RC requires periodic calibration and a warm-up period.

Transporting within the construction site is difficult due to the senses in the devices until the required attention is supplied, particularly on a route road. Working on uneven ground reduces the scanner's field of view, making spatial data collecting more difficult. Because of the overlapping angle of the camera view, generating semantic data is challenging and analyzing structures above the aircraft is tough [48,53,100]. The biggest constraint of RC on construction site monitoring is at the project completion phase, when line-of-sight accessibility is a problem [53]. The acquisition of data needs to ensure that all scanning targets are gathered with the necessary information within time limitations to generate sufficient point cloud data for various uses. To minimize job site activity delays, data gathering efficiency is critical. A previous planning optimization technique known as Planning for Scanning can be used to acquire efficient and effective point clouds data [122]. Laser scanners are a great tool for measuring existing structures, but they do not always work since they suffer from the same issue as visual surveys: if you cannot see it, you cannot scan it, which means the scanner's laser beam will not be able to measure a part of the structure's surface that is not in the scanner's field of view [123]. This constraint applies to the surfaces of roof structures in general, as well as specific features of the façade. In some cases, the generated point cloud may be devoid of data from the building surface, causing problems with the final findings. A suitable alternative could be a mix of two technologies: terrestrial laser scanner and aerial photogrammetry. The practice of acquiring images from an unmanned aerial vehicle (also known as airborne imagery) is known as aerial photogrammetry (i.e., "drones") [123].

### 3.3.4. Human Capacity Building/Professional Training Requirement

Many professionals are still uninformed regarding new technologies, including their benefits, drawbacks, and applications, due to the rapid advancement of technology. Many construction experts are unfamiliar with RC equipment such as laser scanning and photogrammetry. As a result, training is required for the device's operating instructions handbook. As a result, long-term training in device operation and maintenance is essential. It requires sufficient training time for staff on how to use the RC devices and data processing. As a result, professional help is necessary. Laser scanning needs the use of specially trained personnel [55,87]. Hence, awareness has to be created among the people by conducting free seminars and demonstrations to expose the technology to AEC professionals [110].

### 3.3.5. Cultural Concerns

Construction sites are constantly changing, RC's capacity to learn and adapt to new situations is essential. Traditional methods are favored over untrustworthy technologies due to the risk associated with construction, as mistakes can have major financial effects. Because of its fragmented character, reforming the construction industry is difficult [124]. A fruitful conversion from old-style to upcoming models necessitates well-matched designs, managing approaches, labor standards, and site operating methods [125]. Construction operations are carried out during the construction stages and they require multi-point accountability from a variety of project specialties. RC technologies will struggle to be operated efficiently unless these disciplines share mutual interests throughout the project cycle [51,56,123].

### 3.3.6. Data Security

RC cloud-based data security has improved, but cyber crooks can still use it. This is a critical issue because it can have financial ramifications as well as jeopardize building site safety. A computer vision system, for example, could be hijacked to incorrectly identify a construction worker operating at a height. Machine learning (ML) solutions that reduce the exposure of high-level sensitive data will be required by construction companies [124].

### 3.3.7. Institutional Barriers

There is a significant institutional barrier existence in the construction industry due to the perception that technology may replace workers, resulting in increased unemployment rates. RC technology, such as a laser scanner, however, may require a considerable amount of time for setting up and continual supervision and services by expert personnel [87]. In this regard, a construction profession with solid experience in RC, applications, and software is necessary to make an easy transition to implementing RC on building projects [23,119].

### 3.3.8. Sharing of Information

Sharing of information is a challenge in the AEC sector. The intellectual property considerations make it difficult to share information given the standards currently in place [56]. There is no existence of a framework for construction business to follow and also, no policy guidance on how to integrate technologies into the websites [125]. Massive data on-site security, trustworthy storage, efficiency, and interpretation are all major challenges.

## 4. Discussion

RC technologies are the digital data acquisition tools for construction management. The number of opportunities and challenges for RC in construction management has been revealed and conceptualized in Table 1. In the current building process, RC is used to track project progress and analyze quality. For RC technologies, effective site progress monitoring is critical. RC technologies are proven tools for efficiency and are time-saving for disseminating information among different project members. RC provides project participants with real-time digital data that they may share, exchange vital information, and make decisions digitally without using any physical media. It is also recommended for every construction project to adopt RC for quality and defect assessment. Construction safety management can benefit from RC technologies. In recent years, RC has shown to be the most effective and efficient technique for acquiring digital data for active construction projects and digital documentation of existing buildings. Another amazing feature of RC technologies is that it provides a project with parametric model visualization and a virtual walkthrough of the project without having to visit the actual project, giving the impression of being in the real environment. From the design stage to the completion stage, RC has made a significant contribution to the construction project in terms of visualization. Many construction businesses are currently using RC efficiently and more are planning to embrace the knowledges for project visuals, construction verification, and project progress inspection. Though RC technologies appear to be an important instrument in the construction business, they have a number of challenges. There appear to be certain barriers when implementing RC in the building sector. A huge initial cost of RC investment, lack of RC experts and technicians, scan visibility issue, and upkeep of the technology are the biggest limitations for construction firms adopting it into their construction management systems. Many scholars predict that future generations will soon overcome all of the disadvantages and limits. If RC technologies advance in terms of safety, quality, visualization, workforce, and time management, it is certain that it will play a larger role in building projects. Professionals and authorities could apply RC technology to various infrastructural projects to achieve the most out of emerging digital construction management, as well as work around restraints or limitations to achieve the maximum benefit through the application of RC in building projects.

**Table 1.** Summary of Opportunities and Challenges using Reality Capture in Construction Management.

| Reality Capture in Construction Project Management | | |
|---|---|---|
| **Stages** | **Opportunities** | **References** |
| Pre-Construction Phase—Planning and Designing | Data accuracy and reliability | [3–5,13,44–46] |
| | Effectiveness and Efficiency of AEC Professionals | [3,47–51] |
| | Site Condition and Design Visualization | [23,52–55] |
| | Reduction in Project Risk | [56–58] |
| | Prevent Cost Overruns | [27,41,58–63] |
| Construction Phase | Data Accessibility | [3,49,64–68] |
| | Site layout planning in large-scale construction projects | [15,69–72] |
| | Monitoring of Site Progress | [3,11,15,54,74–79] |
| | Enhanced Stakeholders Collaboration | [26,55,80–83] |
| | Remote Construction Supervision and Control | [26,54] |
| | Health and Safety Assessment | [72,84,85,88,89] |
| | Project Team Communication and Data Acquisition | [2,17,26,65,90,91] |
| | Enhance Assessment of Work Done and Approval for Payment | [41,50,92–94] |
| | As-built Model Visualization | [47,52,54,95–97] |
| | RC Gives Life to Building Information Modeling (BIM) | [3,4,13,59,98–102] |
| | Scheduling and Cost Monitoring | [26,54,56] |
| | Productivity Measurement | [103] |
| | Defect and Quality Assessment | [55,64,92,104–110] |
| | As-built Digital Documentation | [23,107,111–115] |
| | Reduction in Variations | [116] |
| **Challenges** | | |
| Pre-Construction and Construction Phase | Higher Initial Investment Costs | [48,54,77,110,112] |
| | Technical Requirement | [23,92,101,119–121] |
| | Field operational difficulties | [48,53,100,122,123] |
| | Human Capacity Building/Professional Training Requirement | [55,87,110] |
| | Cultural Concerns | [51,56,121–123] |
| | Data Security | [124] |
| | Institutional Barriers | [23,87,119] |
| | Sharing of Information | [56,125] |

## 5. Conclusions

Construction is one of the most important industries in the world economy. Huge developments have happened in the construction sector since its inception. RC technologies are bringing about inconceivable changes and advancements in numerous construction management issues, among the many modifications. This research aimed to examine the opportunities and constraints associated with the application of RC technologies in construction project management, as well as its contributions to resolving several difficulties in construction management throughout the last two decades. The study reveals that a remarkable advancement in RC technologies has a significant influence on the construction industry. The benefits and drawbacks of adopting RC technology in construction project management concerns were discussed and summarized in this paper. The identified twenty opportunities were grouped into pre-construction and construction phases with five and fifteen, respectively. Some of the identified opportunities are data accuracy and reliability, AEC professional effectiveness, site condition and design visualization, data accessibility, monitoring of site progress, etc. Eight RC-associated constraints were identified which include high initial investment cost, technical requirement, human capacity building/training requirement, etc. Summarized information promotes education on the enormous advantages RC offers in the area of construction project management to drive its adoption. It might be beneficial for readers to look at the various applications of RC and how projects can benefit from the technology. The future of RC depends on the

existence of systems that address problems of 'large scale' digital construction. More research has demonstrated the importance of offering improved user interference and enabling previously unthinkable applications from both evolutionary and revolutionary perspectives. In addition, the study was limited to Scopus database only, hence further study should consider other databases for a similar search to augment the research area. A similar study to consider post-construction management is currently under review. The authors recommend a separate review on each RC technology, such as the UAV or LIDAR, to appreciate its associate challenges and opportunities regarding the detailed technology application and operation.

**Author Contributions:** G.F. developed the methodological concept, data collection, and analyses and drafted and edited the manuscript. I.M. and F.M. provided supervision, reviewed, and conceptualized the direction of the study. All authors have read and agreed to the published version of the manuscript.

**Funding:** This research is funded by the University of Johannesburg and supported by Kumasi Technical University.

**Institutional Review Board Statement:** The study was conducted in accordance with the University of Johannesburg's Research Ethics Committee.

**Informed Consent Statement:** Informed consent was obtained from all subjects involved in the study.

**Data Availability Statement:** Not applicable.

**Acknowledgments:** The work is supported and part of collaborative research at the Centre of Applied Research and Innovation in the Built Environment (CARINBE), University of Johannesburg. The authors wish to acknowledge the University of Johannesburg for the resources used to conduct this study.

**Conflicts of Interest:** The authors declare no conflict of interest.

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
