# Peer review of "Reality Capture in Construction Project Management: A Review of Opportunities and Challenges"

_buildings, doi:10.3390/buildings12091381_

Round 1

Reviewer 1 Report

The authors have attempted to understand the challenges and opportunities associated with Reality Capture as a systematic literature review. This is an important topic at the right time when the directions of research in this area are being explored. The article starts with the right note. But I have a few significant concerns that I would like the authors to address. I have put down my concerns below:

1. The authors tried to club all the technologies associated with Reality Capture under a single umbrella. We should appreciate that different technologies like the UAV or LIDAR have associated challenges and opportunities. By sweeping the entire range of technologies under one umbrella of RC, I felt the paper stopped at a much broader level and could not go into the detail of which technologies have what challenges etc. 

2. A related query is regarding the findings and discussion section. All these problems are well documented, as the authors prove by conducting this systematic study. What is the new finding from the survey about the literature or knowledge which is interesting to the reader and can be revealed only after reading about 100 papers? Did we get a big picture of the challenges and opportunities available? Right now, all I see is the list of challenges and opportunities which little discussion on what bigger picture was revealed by the systematic review?

3. Coming to the research method, what was the objective and motivation that made the authors go for a review of literature in this area. This is not clear in the manuscript. 

4. PRISMA is a good approach if handled meticulously. My questions here are 

4.a. How were the inclusion criteria decided?

4.b How was the eligibility determined? Is it replicable if a new researcher does this study again?

4.c Why did the authors choose only open access journals. (as indicated by the inclusion criteria in the figure). Please correct me if I was wrong in this assumption but make it clear in the manuscript. 

4.d Why was Scopus only used for the study?

5. Some more understanding of the trends and scientometrics would have been interesting. 

6. The project phases just talked about planning&design and construction. Why not facilities management/rehabilitation phases?

7. How is the reader assured that the article comprehensively captures the issues and opportunities. 

8. I missed a section in the article discussing future research directions and opportunities. This should be included. 

The authors may address the above concerns. 

Reviewer 2 Report

Reality Capture in Construction Project Management: A Review of Opportunities and Challenges

This study uses the PRISMA procedure to conduct a systematic literature review, and articles were considered relevant. The articles were analyzed using Content analysis techniques to synthesize the themes.

Significant output of this study is required to include in abstract with proper justifications.

References are not according to format.

Also, mostly references are old, latest studies are required in introduction.

There is need to further improve section 2.

Section 3.1, proper reasons are not provided for observed trends, for example;

In the 2013, the number of publications showed an upward trend of 9 which indicates that the application of RC is attracting increasing attention from researchers in the AEC industry. Why?

What is Y-axis in Figure 2?

Line 189. To effectively implement novel technologies into projects, professionals need to recognize the benefits and performance enhancements RC  offer to the built environment.

The authors must proposed some solutions for professionals. 

what is objective of Figure 3.

Authors must summarized review results in more systematic way with reference to the previous studies.

Also, Conclusions are too limited to proof the significant outcome of this study.
